# Telomere Length and Telomerase Activity in Subcutaneous and Visceral Adipose Tissues from Obese and Non-Obese Patients with and without Colorectal Cancer

**DOI:** 10.3390/cancers15010273

**Published:** 2022-12-31

**Authors:** Sergio García-Martínez, Daniel González-Gamo, Sofía Elena Tesolato, Ana Barabash, Sofía Cristina de la Serna, Inmaculada Domínguez-Serrano, Jana Dziakova, Daniel Rivera, Antonio José Torres, Pilar Iniesta

**Affiliations:** 1Department of Biochemistry and Molecular Biology, Faculty of Pharmacy, Complutense University, 28040 Madrid, Spain; 2Sanitary Research Institute of San Carlos Hospital (IdISSC), 28040 Madrid, Spain; 3CIBERDEM (Network Biomedical Research Center for Diabetes and Associated Metabolic Diseases), Carlos III Institute of Health, 28029 Madrid, Spain; 4Endocrinology & Nutrition Service, San Carlos Hospital, 28040 Madrid, Spain; 5Department of Medicine, Faculty of Medicine, Complutense University, 28040 Madrid, Spain; 6Digestive Surgery Service, San Carlos Hospital, 28040 Madrid, Spain

**Keywords:** adipose tissues, obesity, telomere, telomerase, colorectal cancer

## Abstract

**Simple Summary:**

The risk associated with obesity for the development of colorectal cancer seems to be well established. However, no biomarkers have been defined that allow the degree of obesity to be related to the clinical evolution of individuals affected by colorectal cancer. With the aim of contributing to the development of this correlation, we carried out a prospective study investigating parameters related to telomere function in the subcutaneous and visceral adipose tissues of a total of 147 subjects affected and not affected by colorectal cancer. Both the relative telomere length and the activity of telomerase in the adipose tissues seem to constitute parameters of interest in the clinical evaluation of individuals with colorectal cancer, which could be also related to the body mass index.

**Abstract:**

To investigate the molecular mechanisms that link obesity and colorectal cancer (CRC), we analyzed parameters related to telomere function in subcutaneous and visceral adipose tissues (SAT and VAT), including subjects with and without CRC, who were classified according to their body mass index (BMI). Adipose tissues were obtained from 147 patients who had undergone surgery. A total of 66 cases corresponded to CRC patients, and 81 subjects were not affected by cancer. Relative telomere length was established by qPCR, and telomerase activity was determined by a method based on the telomeric repeat amplification protocol. Our results indicated longer telomeres in patients affected by CRC, both in SAT and VAT, when compared to the group of subjects without CRC. Tumor local invasion was associated with telomere length (TL) in SAT. Considering the BMI values, significant differences were found in the TL of both adipose tissues between subjects affected by CRC and those without cancer. Overweight subjects showed the greatest differences, with longer telomeres in the group of CRC patients, and a higher number of cases with telomerase reactivation in the VAT of subjects without cancer. In conclusion, parameters related to telomere function in adipose tissue could be considered as potential biomarkers in the evaluation of CRC and obesity.

## 1. Introduction

Telomeres are protective nucleoprotein structures at the end of eukaryote chromosomes that play a key role in the protection against genome instability-promoting events [1]. Telomere shortening has become a widely accepted molecular measurement of aging, and tissue telomere length (TL) represents a strong marker for the aging process [2,3].

In individuals with obesity, differences in fat distribution and adipose tissue cellular composition may contribute to obesity-related metabolic diseases. Significant TL differences between the superficial SAT of lean and obese subjects, with and without type 2 diabetes, have previously been reported [2]. Moreover, obesity is an established risk factor for different cancer types, including CRC. The main biologic mechanisms whereby obesity and other risk factors are related to cancer incidence include several emerging pathways related to oxidative stress and TL, among others [4]. Obesity seems to be related to shorter telomeres, and fat depot differences in TL support the notion that subcutaneous pre-adipocytes have a higher capacity to differentiate than omental pre-adipocytes [2]. Compared to SAT, VAT has revealed higher levels of markers of inflammatory lipid metabolism, some of them associated with CRC tumor stage [5,6].

CRC is one of the most common malignant tumors in the word, representing one of the highest rates of morbidity and mortality worldwide [5,7]. In a previous study, our research group jointly evaluated the prognostic relevance of obesity and telomere status in patients affected by CRC who had undergone surgery with a curative intention. Our results demonstrated that the length of the telomeres is a useful biomarker to predict the clinical outcome in these subjects. Patients with shorter telomeres, both in tumor and their non-tumor paired tissues, had the best clinical evolution, independent of the Dukes’ stage [8,9]. Interestingly, our results indicated significant differences according to gender. Thus, when the effect of obesity on the prognosis of patients with CRC was analyzed, the differences were only evident in the male population [9].

In cancer patients, visceral adiposity has been associated with poorer clinical outcomes, such as postoperative complications, survival, and recurrence, in both the short and long term [10,11]. It has been suggested that the dysregulated deposition of excess adiposity is prognostic of mortality in CRC [12]. However, addressing the impact of body fat in CRC biological behavior is still an unmet need. Understanding how adiposity influences CRC staging and prognosis could allow for further patient risk stratification to devise targeted interventions and improve clinical outcomes [13].

In order to contribute to the detection of biomarkers that could be useful in the clinical evaluation of obese patients affected by CRC, in this work, we tested potential differences in the relative TL and telomerase activity between abdominal SAT and VAT in individuals with and without CRC who were classified regarding their BMI values. A total of 147 patients were included in our analyses, 66 affected by CRC and 81 without CRC.

To our knowledge, no reports regarding changes in parameters related to telomere function in adipose tissues from patients with and without CRC have been previously published.

## 2. Materials and Methods

### 2.1. Patients and Tissue Samples

Paired abdominal SAT and VAT were obtained prospectively from 147 patients who had undergone surgery at San Carlos Hospital in Madrid, Spain, over the last 10 years. A total of 66 subjects (28 women and 38 men) were CRC patients who submitted to potentially curative surgery, whereas the remaining 81 subjects (56 women and 25 men) were not affected by cancer, but had undergone bariatric surgery, and were considered as the control group in the present study. The mean age ± standard error of the study cohort was 63.38 ± 1.19 years (62.53 ± 1.62 for women and 64.43 ± 1.77 for men). Considering only the group of CRC patients, the values for age were 70.04 ± 1.56 years (71.48 ± 2.41 for women and 68.98 ± 2.05 for men), whereas in the control group, the mean age ± standard error was 57.38 ± 1.51 (57.39 ± 1.82 for women and 57.37 ± 2.73 for men). Patients were classified according to their BMI values, following the criteria of the World Health Organization (WHO). Thus, patients with a BMI < 25 kg/m^2^ were considered to have normal weight (*n* = 15, all of them affected by CRC); patients with a BMI ≥ 25 kg/m^2^ and ≤ 29.9 kg/m^2^ were considered as overweight (*n* = 47, 27 of them with CRC, and 20 without CRC); and those with a BMI ≥ 30 kg/m^2^ were defined as obese, (*n* = 85, 24 of them with CRC, and 61 without CRC). Cases were collected independent of gender, age of the patient, or tumor stage, in the case of CRC subjects. Moreover, no CRC patient had received previous chemo- or radiotherapy prior to surgery and inclusion in the study.

After surgical resection, all tissue samples were instantly frozen in liquid nitrogen and stored at −80 °C until processing (collection of samples C.0001253). CRCs were pathologically staged according to the original Dukes’ staging scheme, modified by Turnbull et al. [14]. The location of the tumor, grade of differentiation, and other clinical-pathological features were also recorded.

Written informed consent was obtained from patients prior to the investigation. This study was approved by the ethical committee of the hospital (C.I. 15/180-E FIS, 24/04/2015), assuring the confidentiality of patient data.

### 2.2. Parameters Related to Telomere Function Evaluation

The relative telomere length and the telomerase activity, both in SAT and VAT, from all of the subjects, with and without CRC included in the present work, were evaluated.

Telomere length analysis. Genomic DNA was extracted from frozen adipose tissue samples, according to the Blin and Stafford procedure [15]. Relative TL was quantified using the quantitative PCR method based in the Cawthon procedure [16]. Each of the samples was quantified in triplicate. This technique consists of a relative quantification of the TL of a sample (T/S ratio) that is given by the number of copies of its telomere sequence (T) between the number of copies of a single copy gene (S), with respect to a reference DNA. As a housekeeping gene, we used *RPLP0* (Ribosomal Protein Large P0). Briefly, the procedure consists of performing two amplification reactions per sample, each containing 20 ng of DNA, to which 5 μL of the FastStart Universal SYBR Green Master reaction mix is added (ROX), as well as the corresponding primers for each of them, in 10 μL of final volume. In the case of the telomere sequence amplification reaction, concentrations of 900 nM of the forward primer and 300 nM of the reverse primer were used, and for the amplification of the *RPLP0* gene, 500 nM of the forward primer and 300 nM of the reverse primer were used. For each pair of primers, a standard curve was created using serial dilutions of a pooled human DNA (mean TL 7 Kbp) to calculate the efficiency (E) of the experiment; E = 10 ^ (−1/slope). Next, we proceeded to calculate the relative TL of each sample with respect to the reference DNA. The following primers were used: TELOMERE 5′GGTTTTTGAGGGTGAGGGTGAGGGTGAGGGTGAGGGT3′ (reverse) and 5′TCCCGACTATCCCTATCCCTATCCCTATCCCTATCCCTA 3′ (forward); *RPLP0* 5′CAGCAAGTGGGAAGGTGTAATCC3′ (reverse) and 5′CCCATTCTATCATCAACGGGTACAA3′ (forward).
T/S Ratio = TELOMERE Efficiency^[Ct Reference DNA−Ct sample]RPLP0 Efficiency^[Ct Reference DNA−Ct sample]

The T/S ratio constitutes a relative indicator of TL. Higher values of the T/S ratios are related to longer telomere sequences.

Telomerase activity determination. Telomerase was measured using the telomeric repeat amplification protocol (TRAP)-based telomerase polymerase chain reaction (PCR)-enzyme-linked immunosorbent assay (ELISA) kit, cat. no. 11 854 666 910 (Roche Applied Science), which allowed us to establish a semi-quantitative assay, as previously reported [8,9].

### 2.3. Statistical Analysis

Statistical analyses were performed using the SPSS software package version 22 (SPSS Inc.). Differences between two or more study groups were calculated using parametric tests, in the case of normal data (ANOVA, followed by the Bonferroni test for comparing multiple groups), or non-parametric tests, when there were no normal data (Kruskal–Wallis, when we compared more than two categories, and Mann–Whitney U test, when we compared two categories). The normality of the data was investigated using the Kolmogorov–Smirnov test and the homoscedasticity conditions of the variables used in this study using the Levene’s test for equality of variances. To compare the means of two related variables, the Wilcoxon test was performed. The Chi-squared test was employed to compare categorical variables, and correlations were assessed by the Spearman test; *p* values < 0.05 were considered statistically significant.

## 3. Results

Considering all of the cases, subjects with and without CRC, included in this study, results from the TL evaluation showed that the T/S ratio (mean ± standard error) for VAT was 0.93 ± 0.05 and 0.92 ± 0.05 for SAT, with a positive correlation between both tissues (*p* < 0.001, Spearman’s rho, Figure 1). 

This significant correlation between the T/S ratio in the visceral and subcutaneous adipose tissues was maintained in overweight subjects (Figure 2A) and in obese individuals (Figure 2B). No significant differences were detected in telomere length in relation to subject’s age (*p* = 0.733, Spearman’s rho).

Considering the total of subjects included in this study, significant differences were also found in the relative TL of SAT, in relation to the BMI of patients. Thus, the T/S ratio showed higher values in the group of subjects with normal weight, when compared to those with overweight or obesity (Table 1), *p* = 0.046 (Mann–Whitney U test). These results should be interpreted considering that the “normal weight” group only corresponded to CRC patients, whereas the “overweight or obese” group integrated subjects with and without CRC. When only CRC patients were included in our analyses, the mean ± standard error for the T/S ratios in VAT and SAT were: 1.06 ± 0.19 and 1.27 ± 0.15, respectively, for the normal weight subjects, and 1.09 ± 0.09 and 1.00 ± 0.08, respectively, for the cases with overweight or obesity. These values did not reach statistical significance (*p* = 0.681 for VAT, and *p* = 0.100 for SAT; Mann–Whitney U test).

Moreover, both in VAT and SAT, a higher number of telomerase positive cases were detected in patients with overweight and obesity (Table 2), *p* = 0.010 and *p* = 0.078 for VAT and SAT, respectively (Chi-squared test).

The following data were analyzed considering the two groups of patients included in this work, subjects affected by CRC and subjects without CRC. Our results indicated that patients affected by CRC showed longer telomeres, both in VAT and SAT, when compared to the group of subjects without CRC. In visceral adipose tissues from CRC patients, the T/S ratio (mean ± standard error) was 1.08 ± 0.08, with statistical differences with respect to values in the group of subjects without CRC (0.77 ± 0.06), *p* = 0.007. In SAT, our data revealed, also with significant differences (*p* = 0.003), that the TL (measured as T/S ratios, mean ± standard error) were shorter in subjects without CRC (0.75 ± 0.06), when compared to the group of CRC patients (1.07 ± 0.07).

Both in subjects with and without CRC, a positive significant correlation was detected between the TL of VAT and SAT (Figure 3, *p* < 0.001 and *p* = 0.003), respectively. 

Considering the clinical-pathological variables of CRCs, we detected significant associations between TL in SAT and the local invasion degree of tumors: colorectal tumors with a higher degree of local invasion (T4) were associated with significantly shorter TLs in SATs from affected patients (*p* = 0.037, Table 3).

In relation to BMI values, significant differences were found in the T/S ratio values, both in VAT and SAT, between the group of patients affected by CRC and the group of subjects without cancer. Specifically, overweight subjects showed the greatest differences, with longer telomeres in the group of patients with CRC: values for VAT (mean ± standard error) were 0.72 ± 0.008 in the group without CRC, and 1.06 ± 0.11 in the group with CRC (*p* = 0.023; Mann–Whitney U test); the values for SAT (mean ± standard error) were 0.65 ± 0.09 in the group without CRC, and 1.07 ± 0.12 in the group with CRC, (*p* = 0.012; Mann–Whitney U test).

Next, telomerase activity was analyzed in relation to the BMI of individuals with and without CRC (Table 4). In individuals without CRC, it was observed that VAT recorded a higher number of cases with telomerase reactivation in overweight subjects (61.1%, 11 out of 18 cases), when compared with the group of individuals with obesity (9.5%, 6 cases out of 63), the differences being statistically significant (*p* < 0.001; Chi-squared test). Still in individuals without CRC, a greater reactivation of the telomerase enzyme was also found in the SAT of overweight subjects (33.3%, 6 cases out of 18) compared to these same tissues in obese subjects (3.2%, 2 cases out of 63). These differences were also statistically significant with respect to each other (*p* = 0.001; Chi-squared test). However, as is also shown in Table 4, these differences were not observed when we considered the group of patients affected by CRC. This group included three subgroups of cases based on BMI values, while the control group (subjects without CRC) did not include a subgroup of subjects with normal weight. In any case, in the control group, no significant differences were observed in telomerase reactivation between the overweight and obese patients.

In summary, both VATs and SATs from patients affected by CRC showed longer telomeres, compared to those from individuals without CRC. Considering BMI values, significant differences were found in the TL of the adipose tissues of subjects affected by CRC and those without cancer. Specifically, overweight subjects showed the greatest differences, with longer telomeres in the group of CRC patients. Moreover, in overweight subjects without CRC, we observed that for both SAT and VAT, a higher number of cases with telomerase reactivation was recorded. 

## 4. Discussion

The risk of developing CRC, one of the tumor pathologies with the highest incidence in the world, is closely related to obesity. However, in contrast to the risk for CRC, the impact of adiposity on disease staging and patient survival is less well-established. Understanding the relationship between adiposity and the clinical variables of CRC could allow for further patient risk stratification in order to devise targeted interventions and improve clinical outcomes [13]. In this context, in this work, we considered that telomere function parameters, previously associated with both CRC and obesity, could emerge as potentially useful biomarkers to stratify CRC patients, with and without obesity, in order to apply the most appropriate therapies. To test this hypothesis, we established a study in which we investigated TL and telomerase activity in both subcutaneous and visceral adipose tissues from patients with and without CRC that were classified according to their BMI values. In previous reports, in obese patients, a high correlation between TL in SAT, VAT, and the leukocytes was established [17]. Moreover, the relationship of SAT distribution to leukocyte relative TL was previously investigated [18].

According to our results, in the total population considered in this work, a positive and significant correlation between TL in SAT and VAT was detected. Moreover, these correlations were maintained in individuals showing BMI values higher than 25 kg/m^2^, which corresponded to the groups of individuals showing overweight or obesity. Similar results were previously reported by other authors [17]. 

Telomere lengths in SAT were lower in the groups of obese or overweight individuals, compared to those in the normal weight subjects. Moreno-Navarrete et al. reported a negative correlation between the SAT telomere size and BMI of subjects [19]. In other published studies comparing the relative TL (T/S ratio) of VATs and SATs between obese and non-obese patients, a significant telomere shortening in the first group of cases was observed [20]. More recently, VAT and SAT telomere lengths from individuals who underwent bariatric surgery correlated negatively with the BMI values of the subjects [17]. In overweight and obese individuals, the telomeres of both adipose tissues would suffer a similar effect from the inflammatory and oxidative environment typical of a high BMI, allowing a correlation to be detected between the size of their telomere sequences [21].

Considering patients with and without CRC, correlations in TL between the two types of adipose tissues, SAT and VAT, were maintained in both groups. Telomere lengths were significantly higher in the group of patients affected by CRC and, in relation to their BMI values, overweight subjects showed the greatest differences. From our results, a lack of correlation between TL and telomerase activity, both in SAT and VAT from patients with and without CRC, can be deduced. These data can be explained considering that most human tumors maintain their telomeres expressing telomerase, whereas a lower but significant proportion activate the alternative lengthening of the telomeres (ALT) pathway. However, evidence about the coexistence of ALT and telomerase has been found in both in vivo and in vitro cellular models, making the distinction between telomerase and ALT positive tumors elusive [22]. In addition, long noncoding RNAs that arise from subtelomeric regions are also implicated in telomerase regulation and telomere maintenance [23]. Likewise, our observations may reflect the effects of tumor factors which reach adipose tissues and alter their telomere function, as these factors may be related to the maintenance and elongation of telomeres. Analyses of various cell lines, including some from colorectal carcinomas, showed that cells released exosomes containing telomerase subunits with catalytic activity into the microenvironment [24]. Moreover, a greater quantification of various oncosomes that contain a type of mi-RNA, called miR-1246, which favors the proliferation, migration, and invasion of tumor cells, has also been discovered in the serum of patients with CRC through its interaction with the Wnt/β-catenin or JAK/STATS pathways [25,26,27]. All these molecules, secreted into the environment through oncosomes, could be able to reach distant tissues, such as VATs and SATs, and promote both telomerase reactivation and telomere elongation. This situation would be responsible for the detection of a greater telomere size and telomerase activity in the adipose tissues of subjects affected by CRC. These data are key to maintaining the theory defending the influence of the tumor microenvironment on the biology of the analyzed adipose tissues [24,27]. 

Focusing the analyses on subjects who do not develop CRC, a greater detection of telomerase activity in the VAT and SAT of overweight subjects, with respect to subjects with obesity, would be a consequence of the detrimental effect that obesity has on the replicative capacity of the ASC (adipose-derived stromal/stem) cells responsible for maintaining the cellular homeostasis of the adipose tissue, causing them to activate telomerase to a lesser extent [28]. However, these differences were not significant when we only considered subjects affected by CRC, in agreement with recently published data indicating that the obesity-associated tumor microenvironment provokes a transition in the transcriptomic expression profile of cells derived from the epithelial consensus molecular subtype (CMS2) CRC patients towards a mesenchymal subtype (CMS4) [29]. Recently, Miller et al., in a systematic review, reported that adipose-derived stem cells (ADSCs) from obese patients exhibit a different cellular profile compared with ADSCs taken from non-obese patients [30]. Although there is heterogeneity across the studies included in this work, the trend seems to be that ADSCs from obese patients generally have impaired in vitro function, despite showing an overall increased mesenchymal stem cell (MSC) content. Examples of this impairment include reduced viability and proliferative capacity due to increased senescence, fibrosis, and a decrease in colony forming units (a marker of proliferative health). Although there are no data in the literature detailing differences between overweight and obesity at the molecular level, the publication by Miller et al. would help to explain the differences found in our study regarding telomerase activity in subjects without cancer, showing different BMI values.

In relation to clinical-pathological variables in CRC, we detected significantly shorter telomere lengths in SAT from patients showing tumors with a high degree of local invasion (T4). These results could be related to an increase in the replication rate of the cells that make up the SAT, which would increase the synthesis of cytokines and stimulate the development of cancer [31,32]. In addition, molecules released by tumors can reach the adipose tissue and activate signaling mechanisms, such as the Wnt/β-catenin pathway, related to the inflammation processes [27,33,34]. For this reason, SAT with shorter telomeres (due to a higher rate of cell division) would be related to tumors with greater local invasion.

## 5. Conclusions

This study investigating parameters related to telomere function in SAT and VAT from patients with and without colorectal cancer reveals the relationship between the length of telomeres in these tissues and colorectal cancer. In addition, we found relevant differences in relation to the BMI values of the subjects included in the protocols. Likewise, the length of the telomeres constitutes a parameter that is related to the degree of local invasion of colorectal cancers. Therefore, according to our results, adipose tissue telomere length and telomerase activity could be considered as potential biomarkers in the clinical evaluation of CRC and obesity.

## Figures and Tables

**Figure 1 cancers-15-00273-f001:**
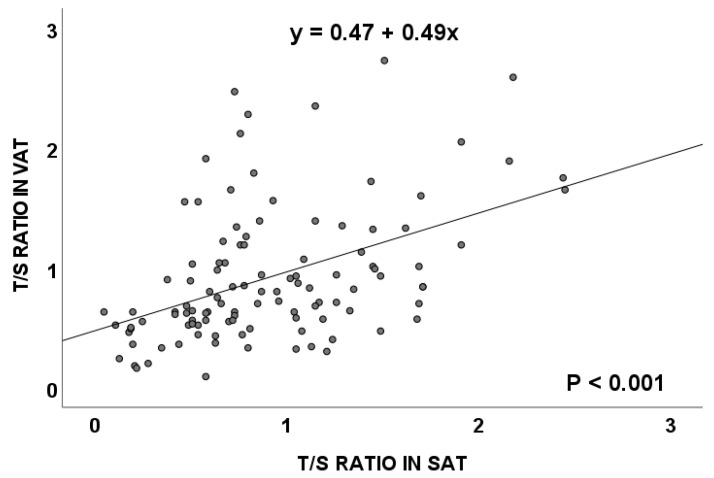
Telomere length (measured as T/S ratio) correlation between visceral and subcutaneous adipose tissues in the total population. VAT: visceral adipose tissue; SAT: subcutaneous adipose tissue.

**Figure 2 cancers-15-00273-f002:**
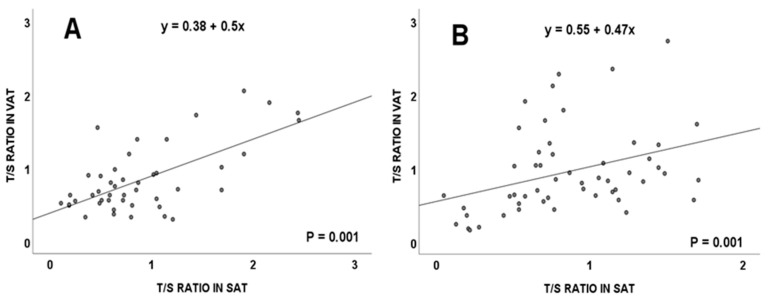
Telomere length (measured as T/S ratio) correlation between visceral and subcutaneous adipose tissues in the total population, considering body mass index from: overweight (**A**) and obese (**B**) subjects. VAT: visceral adipose tissue; SAT: subcutaneous adipose tissue.

**Figure 3 cancers-15-00273-f003:**
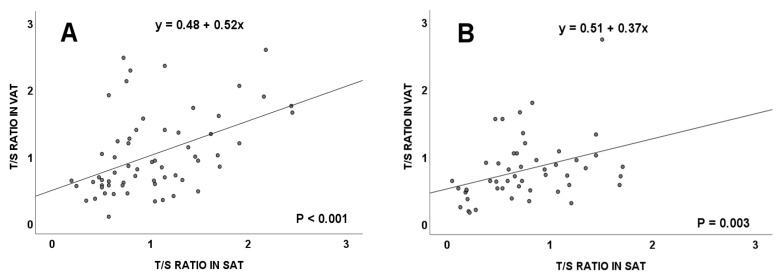
Telomere length (measured as T/S ratio) correlation between visceral and subcutaneous adipose tissues in colorectal cancer patients (**A**), and in subjects without colorectal cancer (**B**). VAT: visceral adipose tissue; SAT: subcutaneous adipose tissue.

**Table 1 cancers-15-00273-t001:** Telomere length, measured as T/S ratio, in visceral and subcutaneous adipose tissues, in relation to the body mass index.

BMI ^3^ Values	Nº of Cases	VAT ^1^ T/S Ratio (Mean ± SE ^4^)	P *	Nº of Cases	SAT ^2^ T/S Ratio (Mean ± SE ^4^)	P *
Normal weight						
(BMI ^3^ ≤ 24.9 kg/m^2^)	15	1.05 ± 0.20		15	1.21 ± 0.16	
Overweight or Obese						
(BMI ^3^ ≥ 25 kg/m^2^)	132	0.91 ± 0.06	0.574	132	0.89 ± 0.05	0.046

^1^ visceral adipose tissues; ^2^ subcutaneous adipose tissues; ^3^ body mass index; ^4^ standard error; ***** (Mann–Whitney U test).

**Table 2 cancers-15-00273-t002:** Telomerase activity in visceral and subcutaneous adipose tissues in relation to the body mass index.

BMI ^3^ Values	Nº of Cases	VAT ^1^ Telomerase Activity		P *	Nº of Cases	SAT ^2^ Telomerase Activity		P *
		Negative	Positive			Negative	Positive	
Normal weight								
(BMI ^3^ ≤ 24.9 kg/m^2^)	15	12	3		15	15	0	
Overweight or Obese								
(BMI ^3^ ≥ 25 kg/m^2^)	132	105	27	0.010	132	117	15	0.078

^1^ visceral adipose tissues; ^2^ subcutaneous adipose tissues; ^3^ body mass index; * Chi-squared test.

**Table 3 cancers-15-00273-t003:** Telomere length, measured as T/S ratio, in visceral and subcutaneous adipose tissues from colorectal cancer patients, in relation to the clinical-pathological variables of the tumors.

Clinical-Pathological Variables of Tumors	Nº of Cases	VAT ^1^ T/S Ratio (Mean ± SE ^3^)	P	Nº of Cases	SAT ^2^ T/S Ratio (Mean ± SE ^3^)	P
Primary Tumor (T)	66			66		
T1	6	1.14 ± 0.38	0.887 ^≠^	6	1.43 ± 0.27	0.037 ^≠^
T2	15	1.00 ± 0.14		15	0.96 ± 0.15	
T3	33	1.11 ± 0.12		33	1.15 ± 0.10	
T4	12	0.98 ± 0.21		12	0.76 ± 0.12	
Lymph Node Invasion (N)	66		0540 ^≠^	66		0.433 ^≠^
N0	42	1.13 ± 0.10		42	1.10 ± 0.10	
N1	16	1.06 ± 0.19		16	0.90 ± 0.08	
N2	8	0.86 ± 0.22		8	1.23 ± 0.22	
Distant Metastasis (M)	66		0.080 *	66		0.343 *
M0	58	1.14 ± 0.09		58	1.10 ± 0.08	
M1	8	0.65 ± 0.12		8	0.86 ± 0.12	

^1^ visceral adipose tissues; ^2^ subcutaneous adipose tissues; ^3^ standard error; ***** Mann–Whitney U test; **^≠^** Kruskal-Wallis test.

**Table 4 cancers-15-00273-t004:** Telomerase activity in visceral and subcutaneous adipose tissues from subjects without cancer (control group) and patients affected by colorectal cancer in relation to the body mass index.

Group of Subjects and BMI ^3^ Values	Nº of Cases	VAT ^1^ Telomerase Activity		P *	Nº of Cases	SAT ^2^ Telomerase Activity		P *
		Negative	Positive			Negative	Positive	
Control group	81	64	17		81	93	8	
Overweight								
(BMI ^3^ 25–29.9 kg/m^2^)	18	7	11		18	12	6	
Obese								
(BMI ^3^ ≥ 30 kg/m^2^)	63	57	6	<0.001	63	61	2	0.001
CRC ^4^ group	66	54	12		66	58	8	
Normal weight				0.962				0.131
(BMI ^3^ < 25 kg/m^2^)	15	12	3		15	15	0	
Overweight								
(BMI ^3^ 25–29.9 kg/m^2^)	27	23	4		27	24	3	
Obese								
(BMI ^3^ ≥ 30 kg/m^2^)	24	19	5		24	19	5	

^1^ visceral adipose tissues; ^2^ subcutaneous adipose tissues; ^3^ body mass index; ***** Chi-squared test; ^4^ colorectal cancer.

## Data Availability

The datasets used and/or analyzed during the current study are available from the corresponding author on reasonable request.

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
