# Peer review of "Telomere Length and Telomerase Activity in Subcutaneous and Visceral Adipose Tissues from Obese and Non-Obese Patients with and without Colorectal Cancer"

_cancers, 2022, doi:10.3390/cancers15010273_

Round 1

Reviewer 1 Report

Here, García-Martínez et al measured telomere length in abdominal subcutaneous and visceral adipose tissues from colorectal cancer patients and non-cancer patients undergoing bariatric surgery. The authors sought to establish a clear correlation between telomere length and cancer stages/prognosis to use it as a predictive measure for colorectal cancer patients. However, the data provided here do not support that telomere length from adipose tissue has any predictive value for the stage or prognosis of colorectal cancer. 

I do not support the publication of this manuscript in its current form. In my opinion, there are major concerns to be addressed and the scientific value of this manuscript is questionable. 

Major questions:

1.     This study is focused on studying the telomere length from different tissues. However, the authors refer to this as “telomere function”, even mentioned in the title. Evaluating the telomere function by checking DNA damage or chromosome stability should be included to keep using “telomere function” throughout the manuscript.

2.     Regarding Table 1: Authors analyzed telomere length between normal weight and overweight/obese patients and claimed that SAT results are significant. This is misleading. Normal-weight patients are only CRC patients, and the other group is a mix of CRC patients and non-CRC patients. These data need to be re-analyzed using normal-weight CRC patients with overweight/obese CRC patients. 

3.     Regarding Table 3: The statistical analysis used in the primary tumor and the distant metastasis is Mann-Whitney U Test, while in the Lymph Node Invasion is Krusbal-Wallis. This should be specified in the Statistical analysis section.

4.     Samples from CRC patients have longer telomeres than non-CRC patients, although CRC samples do not exhibit as much telomerase activity. To explain this contradictory result, the authors mentioned that exosomes from tumors can carry telomerase to other tissues and lead to telomere elongation in extra-tumoral tissues. This hypothesis is not supported by their data. In fact, the authors found more samples positive for telomerase activity from overweight non-CRC patients than in CRC samples. Telomerase activity, even coming from exosomes, should have been observed in CRC samples. How can the authors reconcile these data?

5.     In this study, there is no control tissue for assessing telomere length. Including the telomere length from peripheral blood would support the hypothesis of using adipose tissue for telomere length analysis.

I have also minor comments, but I would like to make a note regarding the writing style. The manuscript is difficult to understand and has several grammatical errors and typos. 

Minor comments:

1.     Within the second paragraph, sentence 10. The authors wrote “tan omental pre-adipocytes” and has to be changed it to “to omental pre-adipocytes”.

2.     VAT has to be explained the first time is mentioned in the main text.

3.     Regarding Table 1: it would help the reader if the telomere length is depicted in CRC patients or non-CRC patients.

4.     Regarding Table 3: The data from the Lymph Node Invasion needs to be adjusted (VAT column), and “Invasión” has to be substituted for “Invasion”.

5.     Regarding Table 3: EE needs to be modified to SE (Standard Error).

6.     Tables 4 and 5 need to be switched due to the order they appear in the text. Table 4 should be Table 5 and vice-versa.

7.     The authors should state a conclusion in the conclusion paragraph. The language in this paragraph is confusing. Mainly, the last sentence is long and difficult to understand.

Reviewer 2 Report

Interesting article with a future application to work on telomere length as a biomarker of cancer and obesity.

In general, the article is understandable and well written. I only comment on some corrections that are marked in the attached pdf document.

Reviewer 3 Report

This is an important paper that proposes telomeres  as promising biomarkers in the clinical evaluation of CRC and obesity

The study is extremely interesting and thorough, but there are some points that require attention.   

1. It is not clear if the authors have compared CRC and noCRC subjects in terms of telomerase activity

2. The absence of the normal-weight group in control subjects is an enormous bias it is necessary to add a sentence to the discussion on this limitation

3. The hypothesis argued by the authors on noCRC group about a greater detection of telomerase both in VAT and SAT of overweight subjects is interesting but is not supported by other studies. This point should be amplied.

4. It would be interesting to analyze a possible mechanism of action that take to the activation of telomerase. For instance the activation of some transcription factors?

Round 2

Reviewer 1 Report

The authors have addressed both major and minor concerns.